# Comparative Chloroplast Genomics and Phylogenetic Analysis of *Persicaria amphibia* (Polygonaceae)

**KyoungSu Choi \*, Yong Hwang and Jeong-Ki Hong**

Plant Research Team, Animal and Plant Research Department, Nakdonggang National Institute of Biological Resources, Sangju 37242, Korea
* Correspondence: choiks010@nnibr.re.kr

**Abstract:** *Persicaria amphibia* (L.) Delarbre, also known as water knotweed, belongs to the Polygonaceae family and has two types: terrestrial and aquatic. We report the chloroplast genome of *P. amphibia* obtained through de novo assembly of Illumina paired-end reads produced by total DNA sequencing. We analyzed the complete chloroplast (cp) genome of *P. amphibia* and found it to be 159,455 bp in length, with a large single-copy region (LSC, 84,281 bp), a small single-copy region (SSC, 13,258 bp), and a pair of inverted repeats (IR, 30,956 bp). It contains 79 protein-coding, 29 tRNA and 4 rRNA genes. Comparative analysis of nine *Persicaria* cp genomes showed a similar genome structure and gene content. However, ycf3 intron II was lost in three *Persicaria* species (*P. hydropiper*, *P. japonica*, and *P. pubescens*) and the SC/IR regions of four species (*P. amphibia*, *P. hydropiper*, *P. japonica*, and *P. pubescens*) included the *rps19* gene. Phylogenetic analysis of the nine *Persicaria* species revealed that *P. amphibia* is sister to *P. hydropiper*, *P. japonica*, and *P. pubescens.* Moreover, we found sequence divergence regions; the largest were *rps16-trnQ*, *trnQ-psbK*, *trnW-trnP*, *ndhF-rpl32*, and *rpl32-trnL* regions. This study could be useful for phylogenetic tree analysis of *Persicaria* and for the identification of *Persicaria* species.

**Keywords:** *Persicaria amphibia*; chloroplast genome; comparative analysis; phylogenetic analysis; nucleotide diversity





## 1. Introduction

*Persicaria* Mill. is a genus in the tribe Persicarieae (family Polygonaceae) that contains approximately 130 species distributed worldwide [1,2]. Based on the macromorphological classification, *Persicaria* is included in the genus *Polygonum* [3]. However, ribulose-bisphosphate carboxylase large subunit (*rbcL*) analysis of Polygonaceae showed that the Polygonoideae subfamily is paraphyletic, and is subdivided into two tribes: Polygoneae and Persicarieae [4].

*Persicaria amphibia*, a species belonging to the genus *Persicaria*, is native to the Northern Hemisphere. It has been described in both aquatic and terrestrial forms, and exhibits complex patterns in leaf morphological variation [5]. *P. amphibia* can grow in aquatic environments as well as terrestrial habitats and has different leaf types. The terrestrial form is erect and has hairy leaves with wavy edges. The aquatic form is hairless with oblong flat leaves. Previous phylogenetic analyses of *Persicaria* used markers of the internal transcribed spacer (ITS) region, and chloroplast DNA (cpDNA) *rbcL* and *trnL-F* [6], which showed that *P. amphibia* formed a clade and was sister to sect. *Echinocaulon* in two cp markers, whereas it was sister to sect. *Tovara* in the ITS marker.

Chloroplasts provide energy to almost all green plants. The chloroplast genome is typically quadripartite in structure, containing a large single copy region (LSC) and a small single copy region (SSC), separated by a pair of inverted repeats (IR) [7] in which organization, gene contents and structures are highly conserved [8]. However, some plants have unique structures owing to gene loss [9–11], rearrangement [12–14], and inversions [15,16].

Recently, cp genome sequences have been used for evolutionary [17,18], taxonomic [19], DNA marker [20,21], and phylogenetic analyses [22–24]. NCBI has uploaded and studied the chloroplast genomes of *Persicaria* species such as *P. chinensis* (NC_050358) [25], *P. pubescens* (MK234901), *P. hydropiper* (MK234902), *P. japonica* (NC_056952), *P. filiformis* (NC_058319), *P. aviculare* (NC_058892), *P. perfoliata* (NC_060649), *P. runcinata* (NC_061176), and *P. maackiana* (NC_061657).

In this study, we sequenced the complete cp genome of *P. amphibia* and compared it with previously published cp genomes of *Persicaria* species. The aims of this study were to: (1) evaluate the phylogenetic position of *P. amphibia*, (2) evaluate the sequence divergence, and (3) suggest useful markers for future phylogenetic studies in *Persicaria* species.

## 2. Materials and Methods

### 2.1. Sampling, DNA Extraction and Sequencing

Fresh leaves of *P. amphibia* were were collected from Gyodongdo, Ganghwa, South Korea, and specimens were deposited in the Herbarium of the Nakdonggang National Institute of Biological Resources (NNIBR). The total DNA content was extracted using the DNeasy Plant Mini Kit (Qiagen Inc., Valencia, CA, USA). Genomic DNA was sequenced using the Illumina Truseq Nano DNA kit (Illumina, San Diego, CA, USA), as per the manufacturer's protocol. Approximately 4.0 Gb of raw data were generated. A total of 31,825,998 reads of the 150 bp paired-end sequence were generated.

### 2.2. Assembly and Annotation

The chloroplast genome was assembled using GetOrganelle [26]. The GeSeq [27] was used to annotate the *P. amphibia* genome. tRNA gene sequences were obtained using tRNAscan-SE [28]. Finally, all published plastid genomes of *Persicaria* were compared using Geneious Prime [29]. OrganellarGenomeDRAW (OGDRAW) was used to draw a circular map of the chloroplast genome of *P. amphibia* [30]. The chloroplast genome of *P. amphibia* was deposited in GenBank (Table 1).

**Table 1.** Comparison of features of nine *Persicaria* cp genomes.

| | *Persicaria amphibia* | *Persicaria pubescens* | *Persicaria hydropiper* | *Persicaria chienesis* | *Persicaria japonica* | *Persicaria filiformis* | *Persicaria perfoliata* | *Persicaria runcinata* | *Persicaria maackiana* |
|---|---|---|---|---|---|---|---|---|---|
| **Length** | | | | | | | | | |
| Total | 159,455 bp | 159,502 bp | 159,054 bp | 159,981 bp | 159,747 bp | 159,741 bp | 160,585 bp | 159,220 bp | 160,595 bp |
| LSC | 84,281 bp | 84,555 bp | 83,835 bp | 84,347 bp | 85,013 bp | 84,432 bp | 85,439 bp | 84,461 bp | 85,376 bp |
| SSC | 13,258 bp | 13,385 bp | 13,357 bp | 12,890 bp | 13,178 bp | 13,073 bp | 12,879 bp | 12,807 bp | 13,055 bp |
| IR | 30,956 bp | 30,781 bp | 30,931 bp | 30,872 bp | 30,778 bp | 31,118 bp | 31,135 bp | 30,884 bp | 31,082 bp |
| **Genes** | | | | | | | | | |
| Total | 113 | 113 | 113 | 113 | 113 | 113 | 113 | 113 | 113 |
| Protein-coding genes | 79 | 79 | 79 | 79 | 79 | 79 | 79 | 79 | 79 |
| tRNA | 29 | 29 | 29 | 29 | 29 | 29 | 29 | 29 | 29 |
| rRNA | 4 | 4 | 4 | 4 | 4 | 4 | 4 | 4 | 4 |
| **GC contents** | 38.2% | 38.2% | 38.2% | 38.0% | 38.1% | 37.8% | 37.5% | 37.9% | 37.9% |
| **Accession number** | This study (ON938209) | MK234901 | MK234902 | NC_050358 | NC_056952 | NC_058319 | NC_060649 | NC_061176 | NC_061657 |

### 2.3. Repeat and Divergence Hotspot Analysis

REPuter was used to visualize forward, palindrome, reverse, and complement sequences with a minimum repeat size of 30 base pairs (bp) and a sequence identity greater than 90% [31]. Simple sequence repeats (SSRs) were detected using MISA [32]. SSRs with minimum numbers of repetitions of 10, 5, 4, 3, 3, 3 for mono-, di-, tri-, tetra-, penta- and hexanucleotides were detected. The alignment of nine *Persicaria* complete chloroplast genome sequences was visually compared using the Shuffle-LAGAN model of mVISATA [33]. The nucleotide diversity (Pi) was determined using DnaSP [34]. The step size was set to 200 bp and the window length was set to 600 bp.

### 2.4. Phylogenetic Analysis

Seventy-seven protein-coding genes of nine *Persicaria* species (Table 1) and one out-group (NC_58892, *Polygonum aviculare*) were compiled into a single file of 66,731 bp and aligned using MAFFT [35]. Maximum likelihood (ML) analyses were conducted using RAxML v.8 [36] with the GTR+GAMMA I model with 1000 bootstrap replications.

## 3. Results

### 3.1. Characteristic of the P. amphibia cp Genome

The complete cp genome of *P. amphibia* (NCBI accession number: ON938209) comprises 159,455 bp with a quadripartite structure and two IRs (30,956 bp) separated by the LSC (84,281 bp) and SSC (13,258 bp) regions (Figure 1, Table 1). The average GC content was 38.2%. The cp genome of *P. amphibia* encoded a total of 113 genes, including 79 protein-coding genes, 29 tRNA genes, and 4 rRNA genes (Table 1). The IR region contained 18 duplicated genes, including seven protein-coding genes (*ycf1, rps7, ndhB, ycf2, rpl23, rpl2,* and *rps19*), seven tRNA genes (*trnN*-GUU, *trnR*-ACG, *trnA*-UGC, *trnI*-GAU, *trnV*-GAC, *trnL*-CAA, and *trnI*-CAU), and four rRNA genes (rrn5, rrn4.5, rrn23, and rrn16).

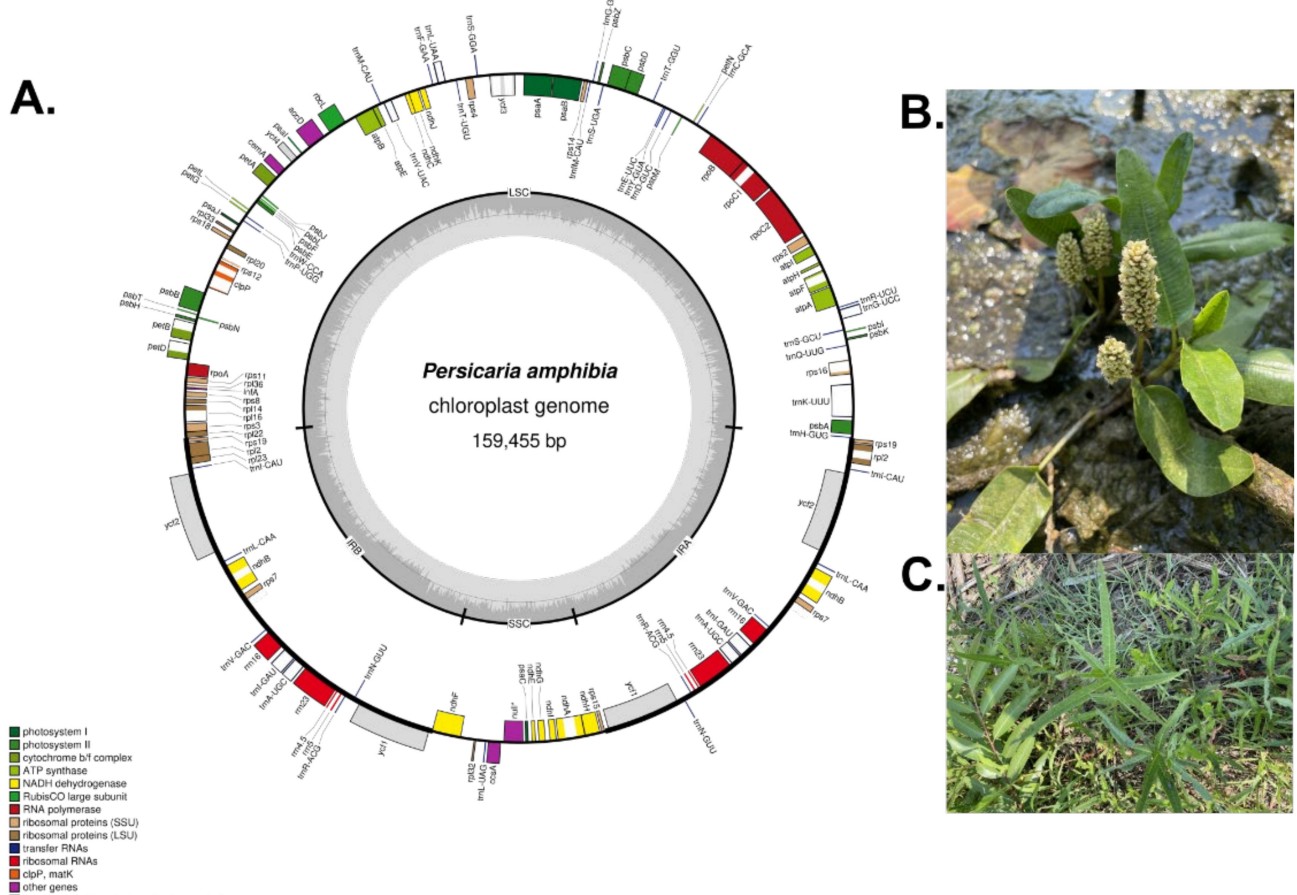

**Figure 1.** The complete chloroplast genome of *P. amphibia* (**A**). Genes drawn inside the circle are transcribed clockwise, and those outside are counterclockwise. The darker gray in the inner circle corresponds to GC contents. Aquatic form of *P. amphibia* (**B**) and terrestrial form of *P. amphibia* (**C**).

Fifteen genes (*rps16, atpF, rpoC1, petB, petD, rpl16, rpl2, ndhB, ndhA, trnA*-UGC, *trnI*-GAU, *trnV*-UAC, *trnL*-UAA, *trnG*-UCC, and *trnK*-UUU) had one intron, whereas three genes (*ycf3, clpP,* and *rps12*) contained two introns.

### 3.2. Comparison of Other Persicaria Species

Nine *Persicaria* cp genomes showed a typical quadripartite structure, consisting of a pair of IRs (30,778–31,135 bp), separated by the LSC (84,281–85,439 bp) and SSC (12,879 13,385 bp) regions (Table 1). The cp genome of *P. aviculare* exhibits the longest genome, and *P. hydropiper* is smaller than other *Persicaria* species (Table 1). Nine *Persicaria* species had the same number of genes (79 protein-coding genes, 30 tRNA, and 4 rRNA genes). However, the second intron of *ycf3* was lost in three species (*P. pubescens*, *P. japonica* and *P. hydropiper* Figure 2).

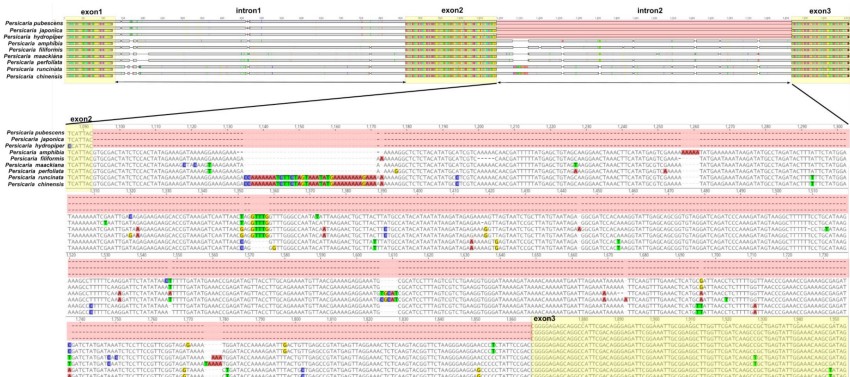

**Figure 2.** The sequences of *ycf3* gene among nine *Persicaria* species. The yellow boxes indicate the exon and red boxes indicate the loss of *ycf3* intron II.

The IR/LSC and IR/SSC junctions of nine *Persicaria* cp genomes were compared. We found two different junction types: IR/LSC and IR/SSC. Type 1 in four species (*P. amphibia*, *P. filiformis*, *P. perfoliate*, and *P. maackiana*) of IR included the intact *rps19* gene. Type 2 in the other *Persicaria* species (except for *P. amphibia*, *P. filiformis*, *P. perfoliate*, and *P. maackiana*) of IR included the partial *rps19* gene. The intact *ycf1* gene was duplicated in IR regions (IRa and IRb). The junctions of IRb and SSC were located in the *ndhF* gene, and the junction of SSC/IRA was located between the *rps15* and *ycf1* genes (Figure 3). However, the total length of IR regions in all of the species was similar (Table 1).

### 3.3. Repeat Sequence Analysis

Simple sequence repeats (SSRs) were detected using MISA [32]. *P. amphibia* had 38 repeats. *P. chinensis* had the highest number of SSRs (59), whereas *P. amphibia* and *P. pubescens* (38) had a smaller number of SSRs than other *Persicaria* species. Most SSRs were A/T mononucleotide repeats (Figure 4A).

Repeats (forward, palindrome, reverse, and complement) were identified using RE-Puter [31]. A total of 25 repeats were present in *P. amphibia* (13 palindrome and 12 forward repeats). The SSRs of other *Persicaria* species were in the range of 19–28. Most of the repeats ranged in size from 30 to 48 bp. *P. japonica* had two long repeats, 154 and 156 (Figure 4B).

### 3.4. Sequences Divergence Hotspots in Persicaria

The cp genomes of nine *Persicaria* species were compared using mVISTA [33]. The results show that protein-coding genes were more conserved than the non-coding regions, and SC regions had lower variation than IR regions (Figure 5).

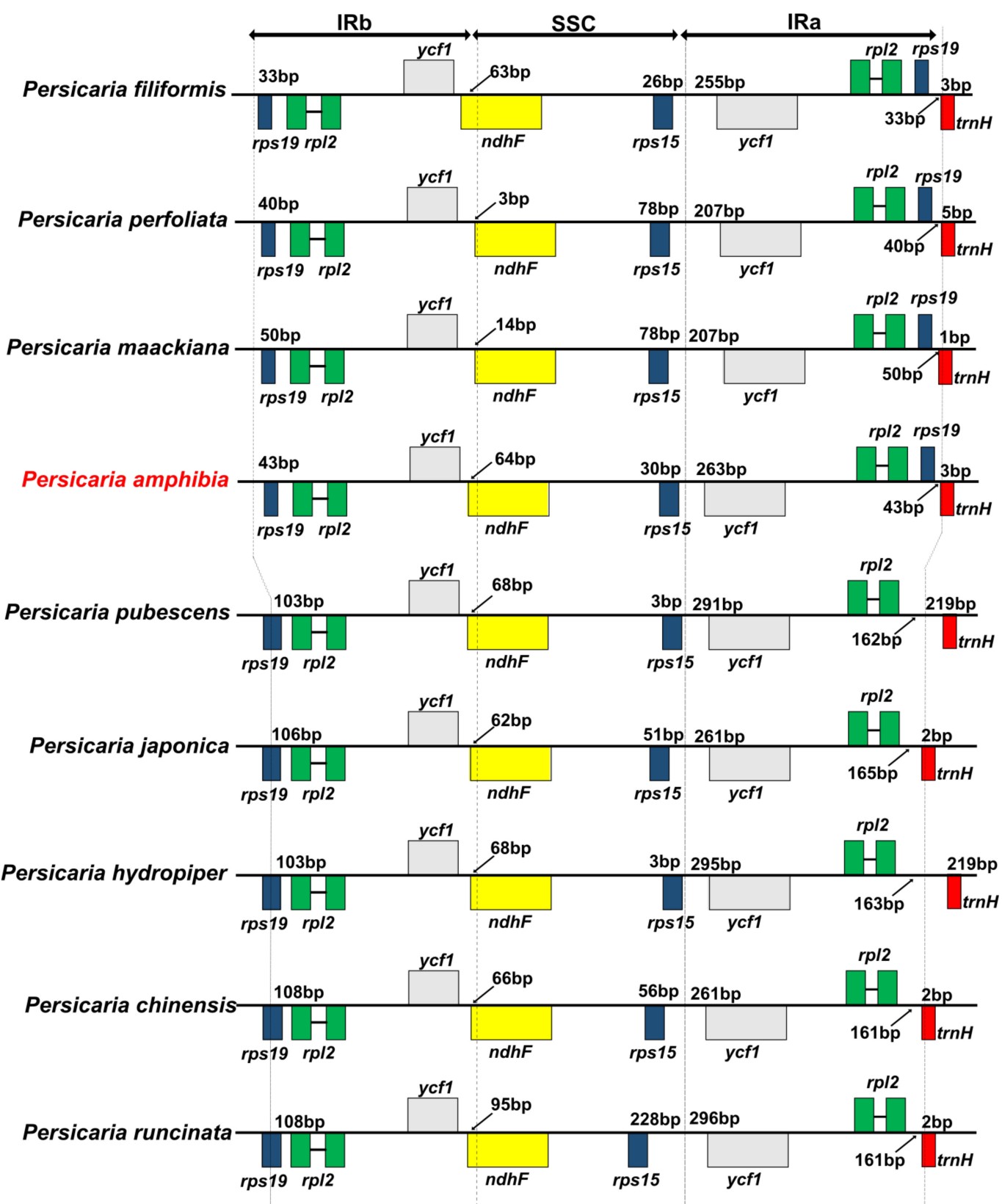

**Figure 3.** Comparison of the LSC/IR and IR/SSC junction among nine *Persicaria* cp genomes.

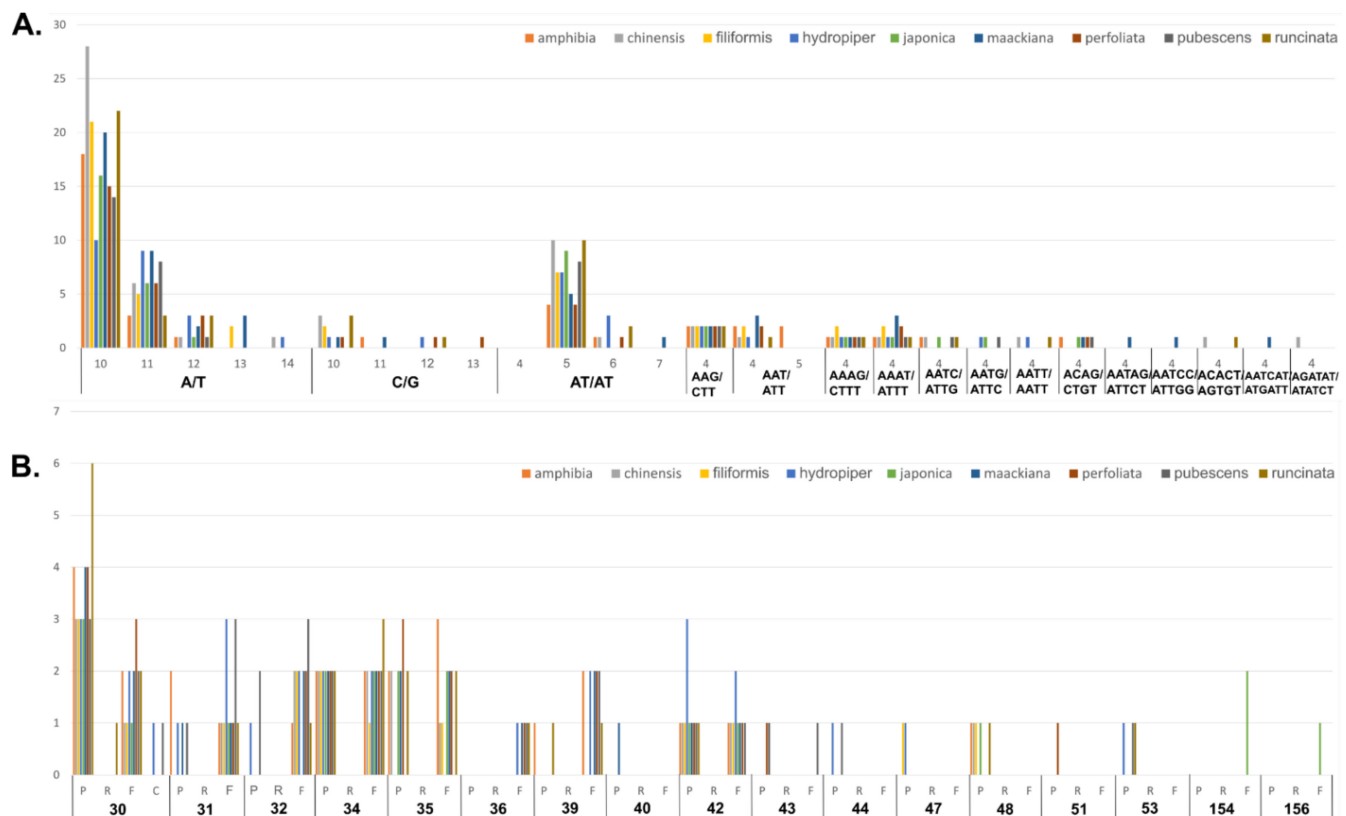

**Figure 4.** Analyses of repeat sequences in the *Persicaria* chloroplast genomes. (**A**) Frequency of SSRs determined by the MISA program. (**B**) Frequency of repeat sequences determined by REPuter.

The number of polymorphic sites and nucleotide diversity were determined using DnaSP [34]. In the complete cp genome, there were 8470 polymorphic sites, and nucleotide diversity (*Pi*) was found to be 0.02746. The *Pi* value was calculated with 200 bp steps. These values ranged from 0 to 0.08875 (Figure 6). The high divergence regions were *rps16-trnQ*, *rpl32-trnL*, *trnW-trnP*, *trnQ-psbK*, and *ndhF-rpl32*, three of which (*rps16*-trnQ, *trnW-trnP*, and *trnQ-psbK*) were located in the LSC region, and two (*rpl32-trnL* and *ndhF-rpl32*) were located in the SSC region. The *rps16-trnQ* region exhibited the highest nucleotide diversity (0.0875, Figure 6).

*3.5. Phylogenetic Analysis*

The ML phylogenetic tree of the nine *Persicaria* species was constructed based on 77 protein-coding genes (66,732 bp, Figure 7). *Persicaria* is monophyletic (BS = 100), and is divided into two clades: (1) *P. runcinata* and *P. chinensis*; and (2) *P. filiformis*, *P. amphibia*, *P. hydropiper*, *P. japonica*, and *P. pubescens*. *P. amphibia* was sister to *P. hydropiper*, *P. japonica*, and *P. pubescens*, with a high bootstrap value (BS = 100).

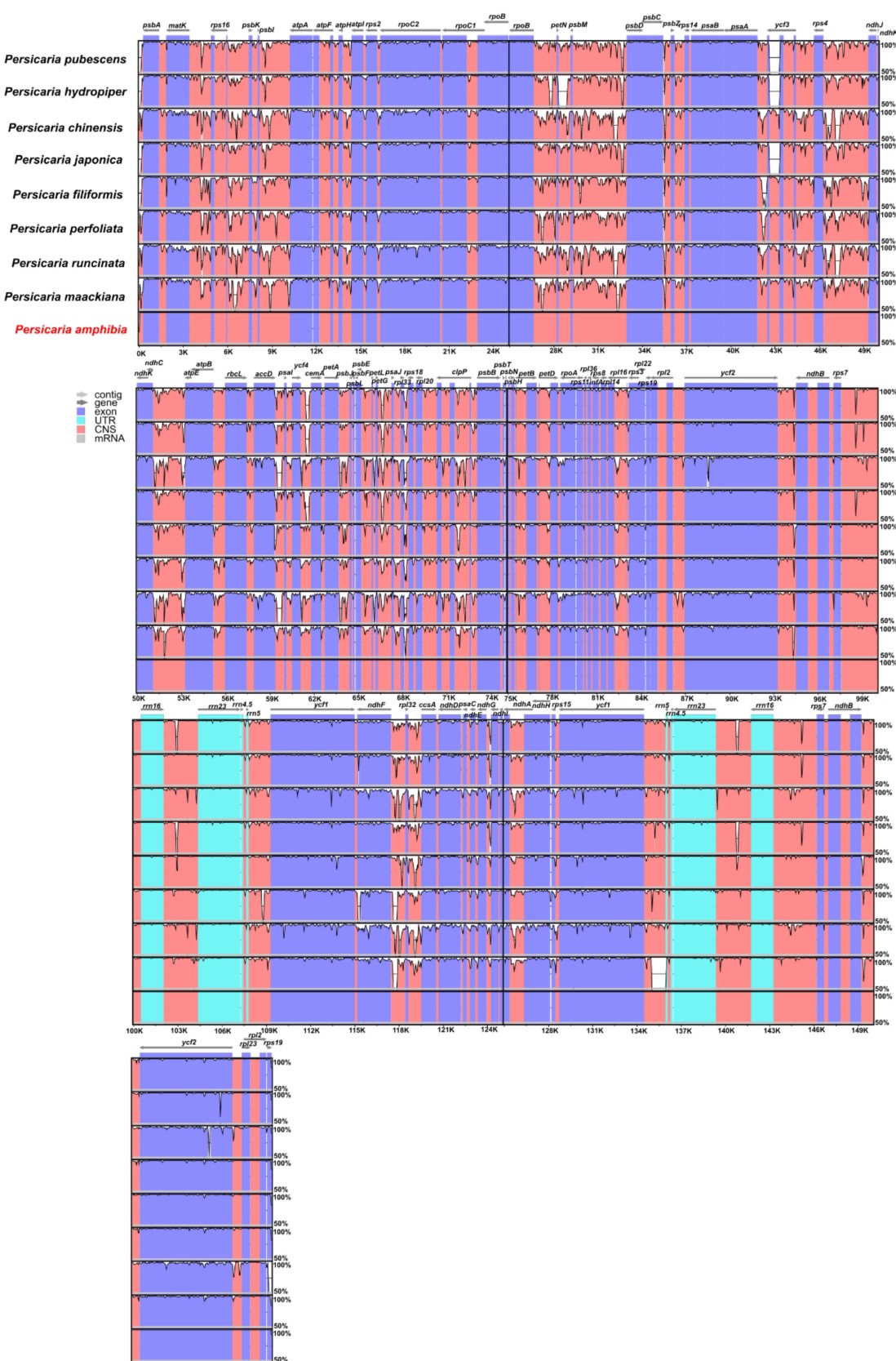

**Figure 5.** Visualization of the alignment of the 9 *Persicaria* chloroplast genomes. The top grey arrow shows genes in order and the position of each gene. The blue and sky boxes indicate protein-coding genes and rRNA.

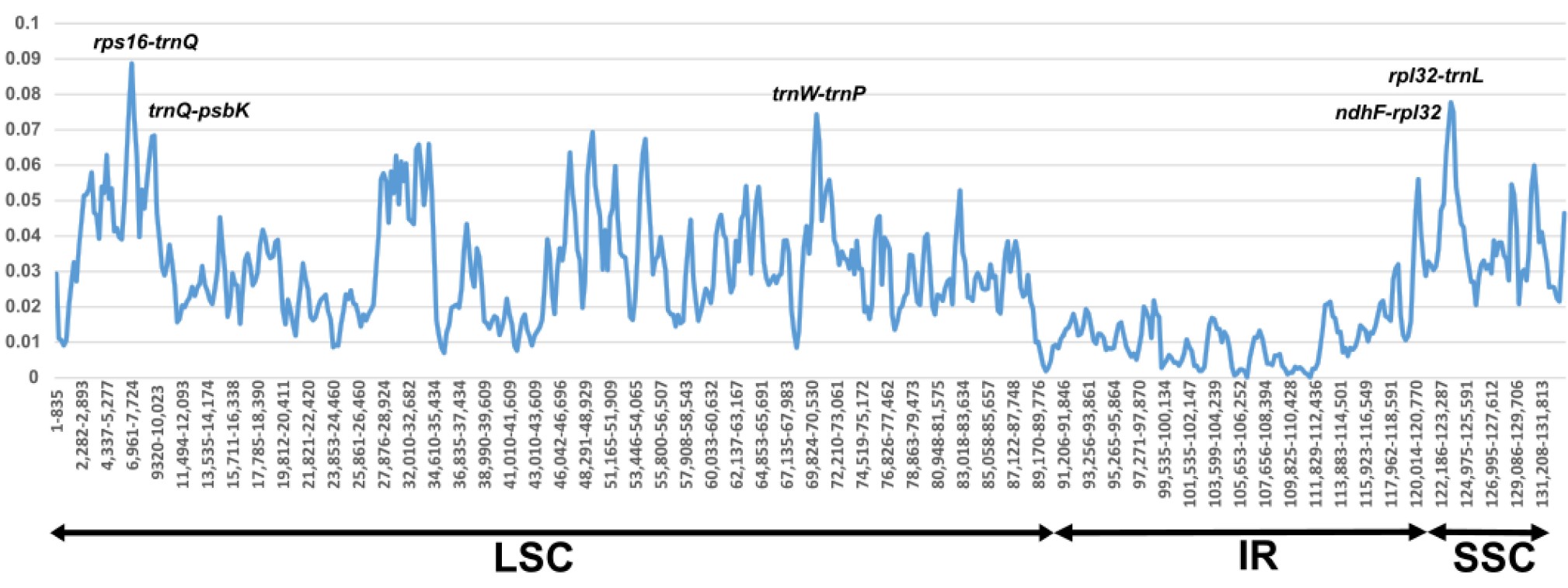

**Figure 6.** Comparison of the nucleotide variability (*Pi*) values in nine *Persicaria* chloroplast genomes.

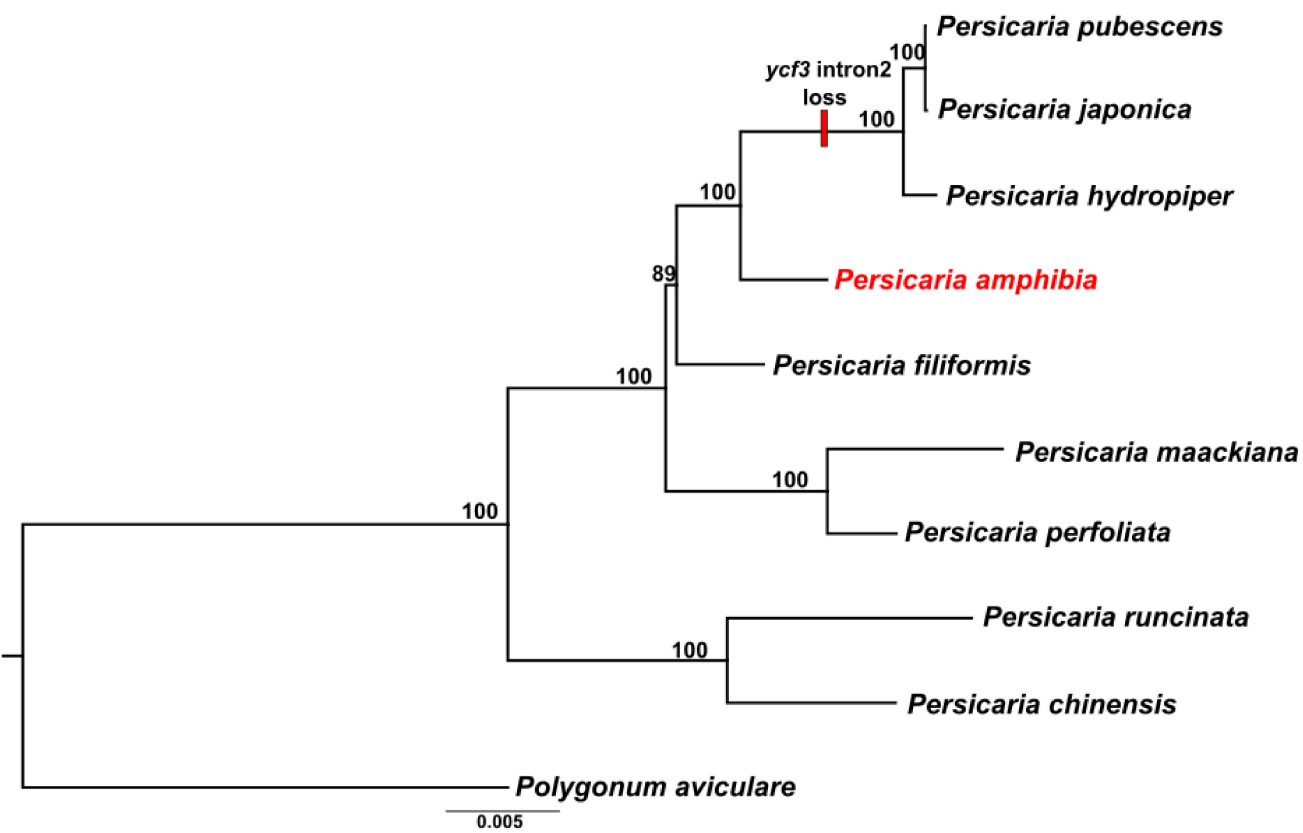

**Figure 7.** Phylogenetic tree reconstruction of 9 *Persicaria* species and one outgroup based on the 77 protein-coding genes (66,732 bp). Number above branches are bootstrap value.

### 4. Discussion

　　Chloroplast genomes are highly conserved in structures and genes. However, previous studies have reported that gene loss in the chloroplast genome occurs in several lineages, including Geraniaceae [37,38], Orobanchaceae [39–42], Fabaceae [43,44], and Pinaceae [45]. Studies have also shown that introns of *clpP* [44,46], *rps12* [46], and *rpl16* [44] are missing in angiosperms. The gene of *ycf3* normally contains three exons and two introns located in the LSC region of the chloroplast genome. The *ycf3* gene encodes photosystem I, a complex in the thylakoid membrane of plants [47–49]. A previous study reported the loss of *ycf3* intron II in *Grammica* and *Cuscuta nitida* [50]. Our results reveal that among the nine *Persicaria* species, three species (*P. pubescens, P. japonica*, and *P. hydropiper)* lost *ycf3* intron II, and *P. amphibia* is sister to these three species. Longevialle et al. [47] suggested that splicing of *ycf3* intron II has an impact on gene *ORGANELL TRANSCRIPT PROCESSING 51* (OPT51) that specifically promotes the splicing of the only group II intron.

　　IR/SC junctions are variable sites in the cp genome, such as IR expansion and contraction [51–54]. Comparison of SC/IR junctions in the nine species showed two types: First, an intact *rps19* gene in the IR region of four species (*P. amphibia*, *P. hydropiper*, *P. japonica*, and *P. pubescens*); and second, a partial *rps19* gene in the IR region of the remaining five species. Previous studies have suggested that partial gene duplication or the contraction and expansion of IR regions causes variable types of *rps19* gene [55–57]. However, the IR regions in the nine *Persicaria* are extremely similar in length (Table 1). This suggests that *rps19* in Persicaria is more likely to be partially duplicated.

　　Previous analysis of cpDNA regions (*rbcL*, *trnL-F*, and *matK*) of *Persicaria* [6] revealed that *P. amphibia* is a sister species to the sect. *Eupersicaria*, such as *P. punctate*, and *P. hydropiper*. However, *P. amphibia* did not show the same results using the nuclear ribosomal DNA (nrDNA) ITS data, indicating that *P. amphibia* is sister to sect. *Tovara* (*P. filiformis* and

*P. virginiana*). Our results show that *P. amphibia* is a sister species to *P. hydropiper*, *P. japonica*, and *P. pubescens*.

*Persicaria* plants are highly variable in their morphological characters. For example, *P. amphibia* has variable morphological characteristics, and some authors have reported two varieties, such as *P. amphibia* var. *emersa* and *P. amphibia* var. *stipulacea*. Additionally, the genus *Persicaria* was reported to exhibit self-fertilization and hybridization [1,58]. Partridge [59] suggested that *P. amphibia* has economic qualities as feed for wild animals, herbal medicine and as an ornamental garden plant. Recently, molecular phylogenetic analysis was used to identify plants [60,61]. Previously, cpDNA (*matK*, *rbcL*) and nrDNA ITS regions have been used for species identification and phylogenetic analyses [52–65]. In *Persicaria*, previous studies have used cpDNA (*matK*, *rbcL*), nrDNA ITS, and *LEAFY* regions. However, the relationships between the genus *Persicaria* are not well supported [1,4–6]. Our results reveal that the highest sequence divergence regions were *rps16-trnQ, trnQ-psbK, trnW-trnP, ndhF-rpl32*, and *rpl32-trnL* in *Persicaria* species. These variable DNA regions could be used as molecular markers and will be helpful in the phylogenetic analysis of *Persicaria* species.

## 5. Conclusions

In this study, we analyzed the complete cp genome sequence of *P. amphibia*. It comprises an LSC (84,281 bp), an SSC (13,258 bp), and two IR regions (30,956 bp), and contains 79 protein-coding genes, 29 tRNA genes, and four rRNA genes. Comparative cp genome analysis of the nine *Persicaria* species showed that all have a similar gene structure and content. However, *ycf3* intron II is lost in *P. hydropiper*, *P. japonica*, and *P. pubescens*. Additionally, we found that the SC/IR junctions in four species (*P. amphibia*, *P. hydropiper*, *P. japonica*, and *P. pubescens*) had an intact *rps19* gene. Phylogenetic relationships based on 77 protein-coding genes showed that *P. amphibia* is sister to *P. hydropiper*, *P. japonica*, and *P. pubescens.* We found five sequence divergence regions (*rps16-trnQ, trnQ-psbK, trnW-trnP, ndhF-rpl32*, and *rpl32-trnL)* that will be useful markers for the further phylogenetic analysis of the genus *Persicaria*.

**Author Contributions:** Conceptualization, K.C. and Y.H.; methodology, K.C.; software, K.C.; validation, K.C., Y.H. and J.-K.H.; formal analysis, K.C., Y.H. and J.-K.H.; investigation, K.C., Y.H. and J.-K.H.; resources, K.C., Y.H. and J.-K.H.; data curation, K.C.; writing—original draft preparation, K.C.; writing—review and editing, K.C., Y.H. and J.-K.H., visualization; K.C., supervision; K.C., project administration; K.C., funding acquisition; Y.H. All authors have read and agreed to the published version of the manuscript.

**Funding:** This work was supported by the grant [NNIBR202201102] from the Nakdonggang National Institute of Biological Resources, funded by the Ministry of Environment, Korea.

**Institutional Review Board Statement:** Not applicable.

**Data Availability Statement:** Not applicable.

**Conflicts of Interest:** The authors declare no conflict of interest.

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
