# Peer review of "Comparative Chloroplast Genomics and Phylogenetic Analysis of Persicaria amphibia (Polygonaceae)"

_diversity, doi:10.3390/d14080641_

Round 1

Reviewer 1 Report

In this manuscript, the authors sequence the whole chloroplast genome of Persicaria amphibia and comparison with other Persicaria taxa. The aims of this study are clear and the results are on good condition. Please check my comments on the attached file and revise them directly. 

Author Response

Thank you for your comment and we look forward to hearing from you. 

Reviewer 2 Report

This article analyzed the complete chloroplast (cp) genome of P. amphibia and found it 159,455 bp in length, with a large single-copy region (LSC, 84,281 bp), a small single-copy region (SSC, 13,258 bp), and a pair of inverted repeats (IR, 30,956 bp). The article suggests useful markers for future phylogenetic studies in Persicaria species. Before recommending this article for publication, there are some shortcomings for that should be resolve.

General comments

Overall, the study is well designed and presented in a good way, but mostly the literature is not cited.  

Abstract

Briefly describe methodology in the abstract.

Add future perspective of the study

Introduction

Provide economic and medicinal importance of the study.

Significance of the phylogenetic must be discussed. The following articles could be cited and take help.

https://doi.org/10.1016/j.jep.2021.114515,

http://dx.doi.org/10.30848/PJB2022-3(19),

Threats to conservation of this species in few sentences.

Typos and English mistakes must be revised to clearly convey the message to readers.

The sentences highlighting previous studies must be cited.  

Author Response

(The authors gave the same response as above.)

Reviewer 3 Report

-The abstract should be slightly rephrased, as in my opinion, it is not very clear that the other cloroplastidial genomes were already published.

-One major issue is that authors state (lines 55-66) that DNA was extracted from fresh leaves of P. persicaria, and that the P. persicaria genome was annotated. Then, in lines 67-68, authors indicate that the P. amphibia chloroplast genome was uploaded to Genbank, which is supposed to be the studied species. Indeed, P. amphibia is mentioned in the abstract, introduction, results and even in the title. Therefore, did they sequence P. persicaria or P. amphibia?

Small comments

Line 10: found 159,455bp

Line 32: should it say erect instead of elect?

Line 55: species name has a typographic error

Line 105-106: please rephrase

Figure 5: it has a very low resolution

Line 216: it comprises a LSC

Author Response

(The authors gave the same response as above.)

Round 2

Reviewer 1 Report

The authors have revised their manuscript according on the reviewers' comments.  I suggest this manuscript can go to the next step to publication.

Reviewer 3 Report

The authors have addressed my suggestions and comments, so I recommend to accept the manuscript in its current form.